

# Are all thermometers equal? A study of three infrared thermometers to detect fever in an African outpatient clinic

Nirmal Ravi[1], Mathura Vithyananthan[2] and Aisha Saidu[1]

[1] EHA Clinics, Kano, Nigeria
[2] eHealth Africa, Washington, D.C., USA

## ABSTRACT

Infrared thermometry has certain advantages over traditional oral thermometry including quick, non-invasive administration and an absence of required consumables. This study compared the performance of tympanic, temporal artery and forehead contactless thermometers with traditional oral electronic thermometer as the reference in measuring temperature in outpatients in a Nigerian secondary care hospital. A convenience sample of 100 male and 100 female adult patients (Mean age = 38.46 years, SD = 16.33 years) were recruited from a secondary care hospital in Kano, Nigeria. Temperature measurements were taken from each patient using the tympanic, temporal artery and contactless thermometers and oral electronic thermometer. Data was analyzed to assess bias and limits using scatterplots and Bland-Altman charts while sensitivity analysis was done using ROC curves. The tympanic and temporal artery thermometers systematically gave higher temperature readings compared to the oral electronic thermometer. The contactless thermometer gave lower readings compared to the oral electronic thermometer. The temporal artery thermometer had the highest sensitivity (88%) and specificity (88%) among the three infrared thermometers. The contactless thermometer showed a low sensitivity of 13% to detect fever greater than 38 °C. Our study shows that replacing oral thermometers with infrared thermometers must be done with caution despite the associated convenience and cost savings.

# INTRODUCTION

## Temperature as a vital sign

Temperature is a vital sign taken during every patient encounter, as fever—generally defined as a temperature above 38 °C—is a sign that the body's normal thermoregulation is altered. The most common reason for fever is a microbial infection of the body. Therefore, body temperature measurements (BTMs) have been instrumental for infectious disease surveillance, as evidenced in the recent epidemics such as SARS, H1N1, Ebola and COVID-19, where there was great need for effective, efficient outbreak monitoring and control (*Bordonaro et al., 2016*; *Ng et al., 2005*; *Plaza et al., 2016*).

Temperature screening at airports was encouraged by west African public health authorities during the 2014–2016 Ebola epidemic to control the transmission of the virus.

Corresponding author
Nirmal Ravi, nirmal.ravi@eha.ng

These temperature screenings allowed the prevention of those who might be febrile from travelling, and thus were part of a co-ordinated attempt to limit the transmission of the virus (*Brown et al., 2014*; *Wickramage, 2019*). This kind of mass transport, community-based, and even at-home temperature screening has also been integral to global containment efforts during the current COVID-19 pandemic (*Jernigan, 2020*; *Wright & Mackowiak, 2020*).

There are several methods of taking body temperature depending on the healthcare setting, patient acuity, healthcare provider partiality, patient preference, accuracy required, and costs involved. Core body temperature can be measured by invasive methods such as esophageal thermometry, pulmonary artery thermometry, and rectal thermometry. Rectal temperature measurement in particular has been seen as the gold standard for accurate temperature measurement (*Allegaert et al., 2014*; *Geijer et al., 2016*; *Sandlin, 2003*; *Yang et al., 2016*). However, it has the disadvantages accompanying invasive procedures including their associated risks, patient discomfort, high costs and chance of infection. Therefore, non-invasive thermometry is the preferred method of measuring patient temperature in most clinical settings. This is also true when attempting to monitor and control infectious diseases in developing countries, where rapid, less-invasive screening processes tend to be favoured by both the public and their policy makers even in non-clinical settings (*Allegaert et al., 2014*; *Hausfater et al., 2008*). Non-invasive thermometry is even more appealing during infectious pandemics as frontline workers can collect temperature readings without physical contact with the patient, thus reducing the risk of disease transmission.

## Non-invasive thermometry

Temperature can be measured non-invasively by methods that require contact or no contact with the body surface. Methods that require contact include oral thermometers, tympanic thermometers, temporal artery thermometers (TAT), and axillary thermometers. *Lawson et al. (2007)* explicitly state that oral measurements are one of the most accurate and precise non-invasive body temperature measurements. However, accurate oral temperature measurements can be influenced by improper probe placement in the mouth by clinicians, as well as the ingestion of hot or cold liquids by patients. Oral thermometry is also contraindicated in unconscious and delirious patients (*Lawson et al., 2007*; *Pappas, 2012*). In addition, because probe covers and frequent alcohol swabs are essential for reducing cross-infection when using oral thermometers, these consumables can add to clinic costs and also add workload to already overwhelmed staff in outbreak-prone areas (*Pappas, 2012*; *Barringer et al., 2011*; *Hayes et al., 2017*).

Tympanic infrared thermometers are noninvasive, inexpensive, quick, and need no consumables. But it can be difficult to position, and have the associated risk of membrane perforation when administered inadequately in both active or sedated patients, and those with ear infections (*Allegaert et al., 2014*; *Sandlin, 2003*). Some studies have shown that tympanic infrared thermometry measurements have increased variability compared to oral and/or rectal measurements (*Allegaert et al., 2014*; *Lawson et al., 2007*). It is also important to be sensitive to patients who may not feel comfortable removing cultural head coverings, and thus preclude adequate access to the tympanic membrane.

Temporal artery thermometers are noninvasive infrared thermometers that measure temperature along the temporal artery on the forehead (*Myny et al., 2005*). TAT has many clinical benefits including the fact that it poses minimal risk of infection, limited risk of injury (*i.e.*, perforation/ discomfort), and it allows for an easily accessible BTM that meets with little patient resistance (*Allegaert et al., 2014*; *Sandlin, 2003*; *Hayes et al., 2017*; *Sandlin, 2003*). Further, many studies comparing the utility and accuracy of TAT in comparison to rectal and oral thermometry showed that TAT can save time for clinicians who work with pediatric populations (*Chiappini et al., 2011*), but also that TAT tends to underperform (*Allegaert et al., 2014*; *Bahorski et al., 2012*; *Reynolds et al., 2014*).

## Forehead contactless infrared thermometry

Among the various infrared thermometry techniques, the one with the least amount of direct risk to patients during measurement is contactless infrared thermometry. In this method, an infrared sensor is placed a few centimeters away from a person's body and the temperature is calculated based on infrared emissions from the body. Such infrared contactless thermometers came into widespread use during the 2014–2016 Ebola outbreak. These are now commonly used in settings as varied as clinics, hospitals, shopping malls, and airports to screen for fever worldwide. Contactless thermometry provides quick, non-invasive temperature measurements without requiring frequent sterilization or consumables  (*Chiappini et al., 2011*). Forehead contactless infrared thermometry is appealing in terms of its low impact on clinician workflow as these thermometers provide quick, non-invasive BTMs that can be easily measured without undressing the patient (*Chiappini et al., 2011*; *Callanan, 2001*; *Carleton et al., 2012*). Though patients and clinicians may show partiality to this non-invasive and contactless BTM method, variability in the reliability and accuracy of forehead contactless infrared thermometry was recently observed (*Berkosy et al., 2018*). In addition, a high false-positive rate of contactless infrared thermometry during mass fever screening in children has been highlighted (*Reynolds et al., 2014*).

## Objective

Infrared contact and contactless thermometers are rapidly gaining use in clinics and hospitals across Africa. There is substantial evidence supporting the use of infrared contact thermometers in clinical settings. However, as a relatively new entrant into thermometry, contactless infrared thermometry does not have a corpus of evidence to support its routine clinical use as a replacement for other established methods. Therefore, the purpose of this study was to compare the accuracy and utility to diagnose fever of three infrared thermometers (tympanic, temporal artery and contactless) against a standard oral digital thermometer in adult outpatients in a Nigerian secondary care hospital.

## MATERIALS AND METHODS

### Participants

A convenience sample of 200 adult outpatients (100 male and 100 female) were recruited over four days in April 2019 from the general outpatient department of a secondary

care hospital in Kano, northern Nigeria. Ethics approvals were obtained from the Health Research Ethics Committee of the Kano State Ministry of Health, Nigeria (MOH/Off/797/T.I/1199, MOH/Off/797/T.I/1208).

Inclusion Criteria:

-Adult patients over the age of 18 who are able and willing to give verbal informed consent to participate in the study.

Exclusion Criteria:

-Patients with altered consciousness

-Patients in distress

-Patients with hemodynamic instability

-Patients with malformation of ears

-Male patients who do not wish to remove their caps

-Female patients who do not wish to remove their head coverings

-Any patient who objects to any of the four methods of temperature measurement

## Apparatus and materials

The following thermometers were used:

- Temporal Artery (TAT 5000, Exergen)
- Contactless (TriTemp, Trimedika)
- Tympanic (Smart Ear, Kinsa)
- Oral digital thermometer (SureTemp Plus 690, Welch Allyn).

## Procedure

### Clinical study

The team leader logged the ambient temperature at the start of the study and every 30 minutes thereafter. Informed verbal consent was obtained and the patient's demographic information was captured using Microsoft Excel®.

If the patient was wearing a headcap or head covering, the patient was asked to remove them for the duration of the temperature measurement. The nurse then waited 5 min before proceeding to wipe the forehead of the patient with a disposable paper towel. The nurse then took the temperature measurements. Only a single measurement was taken per device. All four methods of temperature measurements were done consecutively in the same participant within a span of 5 min. The sequence of the thermometry (Oral → tympanic → temporal artery → contactless) was cycled with each participant so as not to introduce bias. Any patient noted to have a temperature greater than 38 °C was directed to the duty nurse. Each thermometer was cleaned using disinfectant alcohol wipe after each use.

### Statistical analysis

Microsoft Excel and STATA 13 were used for statistical analysis. Statistical significance was set at a $p$-value less than 0.05 and 95% confidence interval. We defined fever as an oral temperature greater than or equal to 38 °C. The degree of agreement of the thermometers and the reference standard was analyzed using Bland Altman plots. Receiver operating characteristics were charted to assess sensitivity, specificity, positive predictive value and

negative predictive value for different thermometry techniques in comparison with oral thermometry. Sensitivity and specificity of the three infrared thermometers to detect fever, as defined by an oral temperature greater than or equal to 38 °C were calculated.

## RESULTS

Half the patients were male and half were female. Ages of the patients ranged between 18 and 82 years (Mean age = 38.46, SD = 16.33). Eight (4%) of the two hundred patients had an oral temperature of 38 °C or higher. The average ambient temperature was 31.5 °C. Figure 1 shows the scatterplots of tympanic, temporal artery and contactless thermometers. Position of the data points in relation to the line of equality (black) gives an indication of the bias of each measurement method. Both tympanic and temporal artery thermometers had similar bias but contactless thermometer had the opposite bias as evident in the scatterplots. True positives (TP), false positives (FP), true negatives (TN), and false negatives (FN) are indicated as the four quadrants created by the intersection of the 38 °C (fever threshold) lines.

We used Bland Altman plots to visualize the agreement of the thermometers and the reference standard (Fig. 2). Bland Altman plots can indicate mean bias and any relationship between the discrepancies and the reference value. The blue dashed lines represent the mean difference in temperature and dotted blue lines represent the 95% confidence interval of the mean difference. The mean difference in temperature measurements between infrared thermometers and oral thermometers, as well as their 95% limits of agreement can be seen in Table 1.

Tympanic and TA thermometers had negative bias of 0.24 and 0.23 respectively compared to the reference thermometer. This signifies that the tympanic and temporal artery thermometers systematically gave higher temperature readings compared to the oral electronic reference thermometer. The contactless thermometer had a positive bias of 0.06, systematically giving lower readings compared to the oral electronic thermometer.

In clinical practice, the ability of a thermometer to accurately detect fever is perhaps more important than its bias compared to a reference standard. We calculated the sensitivity, specificity, negative and positive predictive values of the three infrared thermometers in comparison to the oral thermometer. As shown in Table 2, our study showed good sensitivity and specificity for temporal artery and tympanic infrared thermometers. Temporal artery thermometer had the highest sensitivity (88%) and specificity (88%) among the three infrared thermometers. The contactless thermometer showed a sensitivity of 13% and specificity of 96%. Positive predictive values for all thermometers were low, ranging between 13% and 23% while the negative predictive values ranged between 96% and 99%. Tympanic and temporal artery temperature readings had moderate correlation with oral temperature as indicated by the Spearman correlation coefficient while contactless temperature had very low correlation with oral temperature.

Finally, we plotted the receiver operating characteristic curves for the three thermometers to graphically present the variation in sensitivities and specificities, shown in Fig. 3. An ROC curve plots true positive rate against false positive rate for different diagnostic cut-offs.

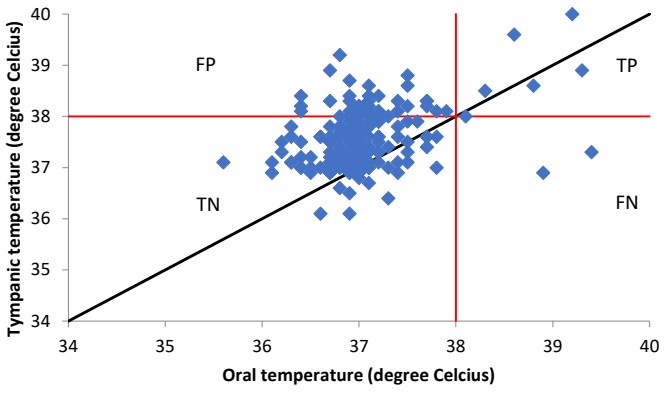

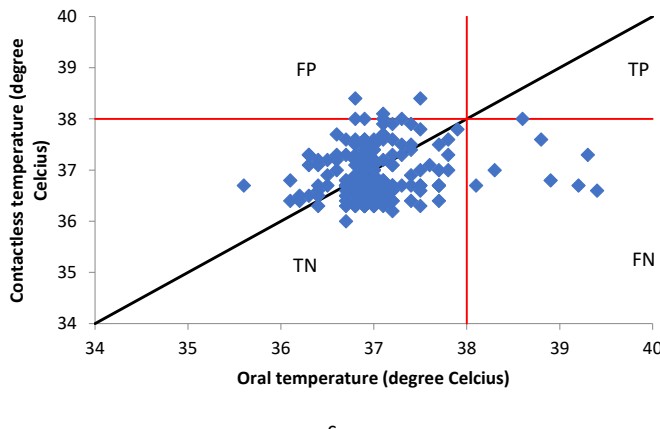

**Figure 1 Scatter plots of tympanic, temporal artery (TA) and contactless thermometers.** (A) Oral temperature *vs* tympanic temperature. (B) Oral temperature *vs* temporal artery (TA) temperature. (C) Oral temperature *vs* forehead contactless temperature.

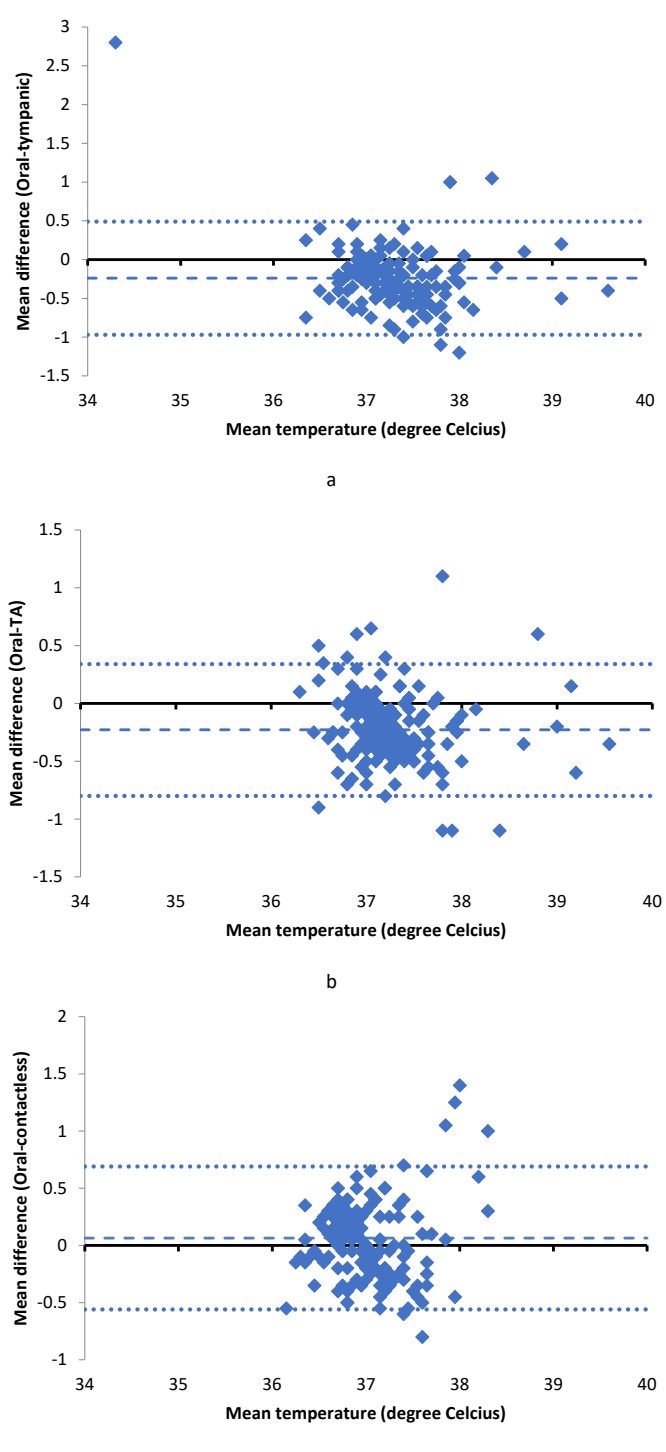

**Figure 2** **Bland-Altman plots of tympanic, temporal artery (TA) and contactless thermometers.** (A) Tympanic thermometer. (B) Temporal artery (TA) thermometer. (C) Forehead contactless thermometer.

**Table 1  Bias of infrared thermometers compared to oral electronic thermometer.**

|  | Tympanic | TA | Contactless |
|---|---|---|---|
| Oral–Infrared bias (°C) | −0.24 | −0.23 | +0.06 |
| 95% limits of agreement of bias (°C) | −0.97 to 0.49 | −0.8 to 0.34 | −0.56 to 0.69 |

**Table 2  Thermometer indices.**

|  | Spearman correlation coefficient | Sensitivity | Specificity | PPV | NPV | ROC AUC |
|---|---|---|---|---|---|---|
| Tympanic | 0.31 | 0.75 | 0.79 | 0.13 | 0.99 | 0.78 |
| TA | 0.28 | 0.88 | 0.88 | 0.23 | 0.99 | 0.87 |
| Contactless | 0.15 | 0.13 | 0.96 | 0.13 | 0.96 | 0.62 |

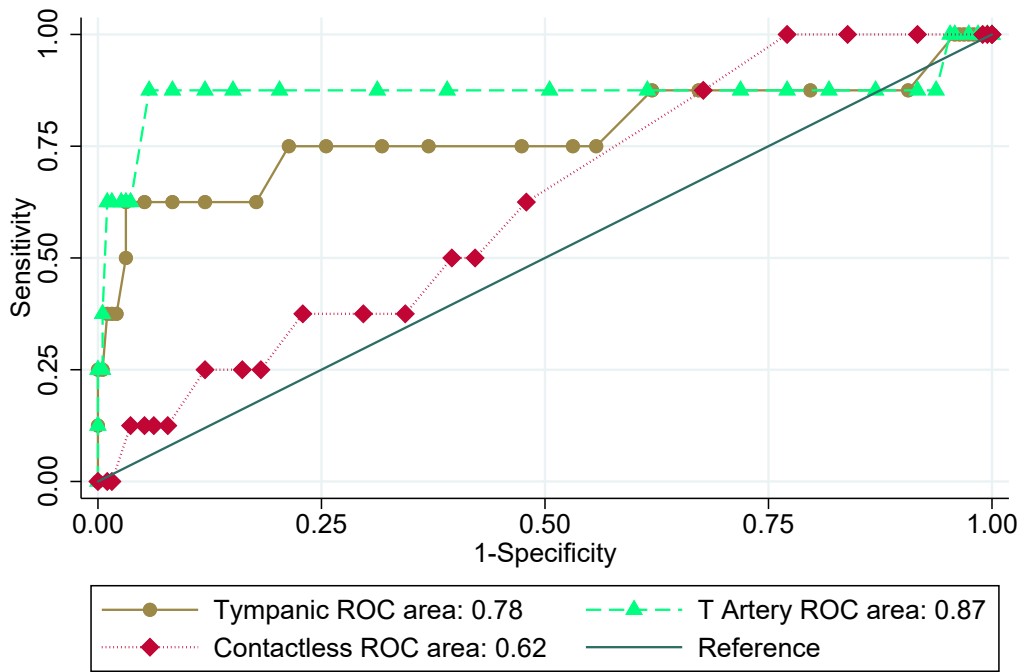

**Figure 3  Receiver operating characteristics of three infrared thermometers.**

Temporal artery thermometer had the highest area under the curve (AUC) of 0.87, followed by tympanic with an AUC of 0.78. The contactless thermometer had an AUC of 0.62.

## DISCUSSION

The goals of our study were to investigate the accuracies of various infrared thermometers, and additionally, to estimate their ability to detect fever in an outpatient clinic setting. We accomplished this by determining the bias of three infrared thermometers in comparison to an oral thermometer and conducting sensitivity analyses. We chose an oral digital thermometer to be the reference thermometer as this has been the standard of care in most

outpatient clinical settings. We wanted to evaluate newer, more convenient thermometers that are relevant in low resource outpatient settings against a standard of care comparator.

## Bias

Scatter plots and Bland Altman charts showed that all three infrared thermometers had bias in comparison to the reference oral thermometer in our study. The Bland Altman chart is a better way to demonstrate bias in measurement methods (*Bland & Altman, 1999*).

Tympanic and temporal artery thermometers had negative bias while the contactless thermometer had a positive bias. The absolute value of the bias was smallest for the contactless thermometer in our study.

A comparison of oral and temporal artery thermometers against esophageal thermometry found smaller but positive bias for the temporal artery thermometer among patients in surgery (*Calonder et al., 2010*). An analysis of axillary and temporal artery thermometer compared to oral thermometer in pre and post operative patients found smaller but negative bias for temporal artery thermometer (*Barringer et al., 2011*). A comparison of contactless, tympanic and temporal artery thermometer with reference to rectal thermometer in pediatric inpatients found no bias for temporal artery thermometer, and positive bias for tympanic as well as contactless thermometer (*Allegaert et al., 2014*). Comparison of tympanic and temporal artery thermometers with bladder reference thermometers showed smaller and a negative bias for temporal artery thermometers among postoperative patients (*Myny et al., 2005*). Differing results between these studies indicate that bias is likely dependent not only on the type of reference device and make/model of index device, but also on the patient population and the clinical setting.

## Correlation coefficients

We found low to moderate correlation between the infrared thermometer readings and oral thermometer readings as indicated by Spearman correlation coefficients. The lowest correlation coefficient of 0.15 was for the contactless thermometer. A comparison of rectal and temporal artery temperature among children under three years of age at a hospital reported Spearman correlation coefficient of 0.86 (*Bahorski et al., 2012*). Spearman correlation coefficient for tympanic thermometer among hospitalized adult patients was 0.93 when compared to nasopharyngeal reference thermometer (*Asadian et al., 2016*). Lin's concordance correlation coefficient was 0.53 for temporal artery and 0.34 for tympanic thermometers among postoperative patients (*Langham et al., 2009*).

## Sensitivity analysis

In an outpatient clinical setting, a thermometer is primarily used to test for the absence or presence of fever. The ability to accurately detect fever is indicated by the positive and negative predictive values of a thermometer. The predictive value of a thermometer is in turn determined by its sensitivity and specificity, as well as the prevalence of fever in the patient population. An ideal diagnostic device will have sensitivity and specificity of 100%, meaning it will correctly identify every positive and negative case. But in reality, sensitivity and specificity of a diagnostic device are often a trade-off with each other. As the sensitivity increases, the device will correctly identify every positive case, but often sacrifice specificity,

which is the ability to correctly identify every negative case. As sensitivity and specificity are fixed for a particular diagnostic device, the positive predictive value increases and negative predictive decreases as prevalence increases.

The prevalence of fever in our population was 4%. Sensitivity was highest for the temporal artery thermometer (88%), while specificity was highest for the contactless thermometer (96%). Contactless thermometer had a very low sensitivity of 13%. This means that the contactless thermometer would only detect 13 out of 100 patients with fever.

Negative predictive value was more than 95% while positive predictive value was lower than 25% for all thermometers in our study population. The negative predictive value is arguably the most important clinical performance characteristic of any diagnostic device used in disease screening. Failing to diagnose fever in febrile patients can cause adverse outcomes such as worsening of disease severity, spreading of the infection to others, higher costs of eventual treatment and possibly even death. These adverse events are more likely in low-resource and rural settings where access to treatment is limited. For a hypothetical fever prevalence of 20%—as can happen in an infectious disease epidemic or a hospital inpatient unit—the negative predictive value of the contactless thermometer would drop to an unacceptable 80%, missing almost one in every five febrile patients. The tympanic and temporal artery thermometers would maintain their negative predictive values of more than 93% even with a fever prevalence of 20%.

A comparison of tympanic, contactless and temporal artery thermometers in pediatric inpatients found sensitivities of 22, 27 and 44 respectively, while the negative predictive values for fever were 94%, 92% and 96% respectively (*Allegaert et al., 2014*). In our study however, tympanic, contactless and temporal artery thermometers had sensitivities of 75%, 13% and 88% respectively, while the negative predictive values were 99%, 96% and 99% respectively. In another study, temporal artery thermometer had sensitivity, specificity and negative predictive value of 83%, 86% and 97% among infants in an emergency department (*Callanan, 2001*). The temporal artery thermometer used in our study had superior sensitivity, specificity and negative predictive value of 88%, 88% and 99%. As with bias discussed above, sensitivity and specificity are inherent characteristics of a diagnostic device, and can vary significantly between different makes, models and the underlying technology. Negative and positive predictive values depend also on the disease prevalence in addition to the inherent device sensitivity and specificity, and can be different for the same device in different patient populations.

ROC curves can be useful to determine the overall accuracy of a diagnostic device. Higher area under the ROC curve is preferred, with an ideal diagnostic device having an area under the curve of 1. We saw the best overall accuracy for the temporal artery thermometer with an AUC of 0.87, while the least accurate was the contactless thermometer with an AUC of 0.62. For context, tossing an unbiased coin as a diagnostic device to diagnose fever would give an AUC of 0.5.

One image that came to define the 2014–2016 Ebola outbreak in west Africa was that of a contactless infrared thermometer pointed at a patient's forehead. Containment efforts of the virus depended on the conspicuous visibility of its incredibly severe symptoms and

its transmissibility only from those who were visibly ill. On the other hand, the highly transmissible COVID-19 traveled the world less conspicuously, and thus necessitated temperature screenings in non-clinical spaces like airports, stores, and even restaurants. It is debatable how much these temperature screenings help with limiting the spread of infectious disease outbreaks.

Notwithstanding, an increasing number of clinics and hospitals are choosing to switch from traditional thermometry to infrared thermometry. Though patients and clinicians may show partiality to this non-invasive and contactless BTM method, our study showed that the forehead contactless thermometer had very poor sensitivity to detect fever. Therefore, if in common use, contactless infrared thermometers may actually result in large numbers of febrile patients being underdiagnosed. Further studies are warranted to determine the precise cut-off temperatures for various thermometers in order to minimize the chances of false negative readings when screening for fever. Considerations must be made to balance accuracy, patient comfort, clinician efficiency and administrative costs. Additionally, considering the limited resources and operating budgets, it would be beneficial to evaluate the cost implications when choosing a particular mode of thermometry in a low-resource clinic or hospital setting. Our study recommends that replacing oral thermometers with infrared thermometers must be done with caution despite the associated convenience and cost savings.

## ACKNOWLEDGEMENTS

The authors acknowledge Dr. Bipin Rajendran for his valuable suggestions regarding data analysis. We also acknowledge the valuable help provided by nurse Musa Abdulmutalab in conducting the study.

### Funding
The authors received no funding for this study.

### Competing Interests
Nirmal Ravi and Aisha Saidu are employees of EHA Clinics. Mathura Vithyananthan was an intern at eHealth Africa.

### Author Contributions
- Nirmal Ravi conceived and designed the experiments, analyzed the data, prepared figures and/or tables, authored or reviewed drafts of the article, and approved the final draft.
- Mathura Vithyananthan conceived and designed the experiments, prepared figures and/or tables, authored or reviewed drafts of the article, and approved the final draft.
- Aisha Saidu performed the experiments, authored or reviewed drafts of the article, and approved the final draft.

## Human Ethics

The following information was supplied relating to ethical approvals (i.e., approving body and any reference numbers):

Kano state Ethics Review Board.

## Data Availability

The analysis code and raw data are available in the Supplemental Files.

## Supplemental Information

Supplemental information for this article can be found online at http://dx.doi.org/10.7717/peerj.13283#supplemental-information.

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
