# Peer review of "Are all thermometers equal? A study of three infrared thermometers to detect fever in an African outpatient clinic"

_PeerJ, doi:10.7717/peerj.13283_

## Round 0.1 · original submission · Minor Revisions

Thank you for your patience. The reviewers were broadly enthusiastic about your work and had very minor comments/questions. Please address these as best as possible.

In particular, and as alluded to by reviewer 2, it would be nice to see a more direct comparison to the results of the other studies you mention. Specifically, lines 383-394 discuss the results of some other studies and it would be great to see a more direct comparison of the types of results that are coming from these studies (and yours). It doesn't look like there is great agreement between different studies, but it would still be nice to see this comparison and the amount of variability that exists when trying to make these comparisons.

·

Basic reporting

Excellent reporting--clear data and methods. Results and discussion are well-informed and articulated.

Minor typo on line 300-301, the word "Oral" has a space or a return between the "r" and the "a"

Experimental design

No comment

Validity of the findings

No comment.

Additional comments

The goal of this paper is to compare traditional oral thermometry with several newer methods adopted more globally during the COVID-19 pandemic. The article is written and organized in an approachable manner that will certainly serve the scientific and medical community, but also the lay person who reads this. Critically, the way this article is written and the data presented are in such a way that the average non-science reader can read and interpret the results.

Reviewer 2 ·

Basic reporting

1. The abstract should clearly state the oral thermometer is used as the gold standard.
2. Line 101: Citations should appear in order (i.e., the first citation to appear in the text should be 1)
3. Line 244: The classification target of fever vs no fever should be explained before describing the metrics used (sensitivity, specificity…)
4. Line 265: Use of Bland Altman plots is described. This should be moved to the Methods section.

Experimental design

The research question is well defined and meaningful. The methods explain the experimental design well, with enough detail to replicate.

Validity of the findings

1. The authors should be commended for undertaking a rigorous comparison of various thermometers, which is a relevant issue given the global pandemic.
2. The authors also do an excellent job of reviewing and summarizing the relevant literature. The authors comment on the literature for bias, but not for correlation and sensitivity. The discussion could be improved by similarly commenting on these metrics and comparing/contrasting with the authors’ own results.

Additional comments

no comment

·

Basic reporting

The authors provide an interesting research article. They use English throughout. But spelling mistakes in the result section ("Spearmann") and the abbreviation of body temperature measurement as BMT instead of BTM, which is used later in the article, should be corrected. The authors provide background information and relevant literature references. The structure of the article conforms to PeerJ standards.

Raw data is supplied and the findings and figures can be reproduced. The definition of fever as a body temperature greater than or equal to 38°C as stated in the statistical analysis section should be used throughout the other sections. The figures are relevant, well labeled and support the content.

Experimental design

The research question is well defined, relevant and meaningful and within the scope of the journal. The results help to narrow the identified knowledge gap. The investigations are performed according to technical and ethical standards. The methods are described with sufficient detail to allow replication of the findings.

Validity of the findings

Demographic data is not given on individual patient level. Apart from that the underlying data has been provided. The conclusions are well stated, linked to the research question and supported by the results.

---

## Round 0.2 · accepted · Accept

Thank you for addressing the reviewer concerns. Congratulations again!